# The Oncogenic Potential of the Centromeric Border Protein FAM84B of the 8q24.21 Gene Desert

**DOI:** 10.3390/genes11030312

**Published:** 2020-03-15

**Authors:** Yan Gu, Xiaozeng Lin, Anil Kapoor, Mathilda Jing Chow, Yanzhi Jiang, Kuncheng Zhao, Damu Tang

**Affiliations:** 1Urological Cancer Center for Research and Innovation (UCCRI), St Joseph’s Hospital, Hamilton, ON L8N 4A6, Canada; yangu0220@gmail.com (Y.G.); linx36@mcmaster.ca (X.L.); mathildachow1994@gmail.com (M.J.C.); xyz989@126.com (Y.J.); kunchengzhao@icloud.com (K.Z.); 2Department of Surgery, McMaster University, Hamilton, ON L8S 4K1, Canada; akapoor@mcmaster.ca; 3The Research Institute of St Joe’s Hamilton, St Joseph’s Hospital, Hamilton, ON L8N 4A6, Canada; 4Department of Medicine, McMaster University, Hamilton, ON L8S 4K1, Canada

**Keywords:** *FAM84B*, *Myc*, 8q24.21 gene desert, H-Ras-like suppressor (HRASLS), Ras, tumorigenesis

## Abstract

*FAM84B* is a risk gene in breast and prostate cancers. Its upregulation is associated with poor prognosis of prostate cancer, breast cancer, and esophageal squamous cell carcinoma. FAM84B facilitates cancer cell proliferation and invasion in vitro, and xenograft growth in vivo. The *FAM84B* and *Myc* genes border a 1.2 Mb gene desert at 8q24.21. Co-amplification of both occurs in 20 cancer types. Mice deficient of a 430 Kb fragment within the 1.2 Mb gene desert have downregulated *FAM84B* and *Myc* expressions concurrent with reduced breast cancer growth. Intriguingly, Myc works in partnership with other oncogenes, including Ras. FAM84B shares similarities with the H-Ras-like suppressor (HRASLS) family over their typical LRAT (lecithin:retinal acyltransferase) domain. This domain contains a catalytic triad, H23, H35, and C113, which constitutes the phospholipase A_1/2_ and O-acyltransferase activities of HRASLS1-5. These enzymatic activities underlie their suppression of Ras. FAM84B conserves H23 and H35 but not C113 with both histidine residues residing within a highly conserved motif that FAM84B shares with HRASLS1-5. Deletion of this motif abolishes FAM84B oncogenic activities. These properties suggest a collaboration of FAM84B with Myc, consistent with the role of the gene desert in strengthening Myc functions. Here, we will discuss recent research on *FAM84B*-derived oncogenic potential.

## 1. Introduction

Tumorigenesis is a complex pathological process. It is affected by sophisticated genetic networks and even more complex epigenetic modifications. The multiplex nature of oncogenesis underlies our continuous effort in the search for cancer etiology. One of the classical genetic events of oncogenesis is amplification of the *Myc* (*c-Myc*) oncogene [1,2]. Myc is the most commonly amplified oncogene across all cancer types [3]. Despite its powerful oncogenic nature, Myc’s oncogenic potential cannot be fulfilled without direct contributions from other oncogenes. For instance, BMI1 (B lymphoma Mo-MLV insertion region 1 homolog) was identified during screenings for potential collaborators for c-Myc-initiated leukemogenesis [4,5]. c-Myc mediates *BMI1* gene transcription to ensure BMI1 availability during oncogenesis for leukemia, neuroblastoma, and nasopharyngeal carcinoma [6,7,8]. Among numerous Myc collaborators, Ras is arguably the most classic one. Their collaboration results in transformation of primary fibroblasts [9] and the activation of cyclin D- and E-dependent kinases [10,11].

The *Myc* gene resides on 8q24.21 and is surrounded by regions known as “gene deserts” as they lack protein coding genes. Downstream (telomeric end) of *Myc* is the *PVT1* gene encoding for a long non-coding RNA (lnRNA), and on its upstream or centromeric side sits a 1.2 Mb gene desert with the centromeric side bordered by the *FAM84B* or *LRATD2* gene (Figure 1). The unique feature of this gene desert is the existence of multiple (lnRNAs) (PCAT1, PCAT2, POU5F1B, CCAT1, CCAT2, CASC8, CASC11, CASC19, and CASC21) with *FAM84B* and *Myc* being the only protein coding genes (Figure 1) [12,13,14]. In view of Myc being the most-well-studied oncogene and the 8q24 gene desert as a region that is frequently amplified in cancer, FAM84B stands as a promising target for oncogenic activities; nonetheless, its impact on tumorigenesis remained unknown until recently. As the oncogenic potential of the non-coding RNAs of PVT1 and those within the gene desert (Figure 1) has been recently reviewed [13,14,15,16,17], we will focus on the emerging role of FAM84B in tumorigenesis in this review. We will briefly discuss the 8q24 gene desert with respect to oncogenesis to set the stage for the following systemic examination of the evidence pertinent to FAM84B-derived tumorigenesis. The main materials used in this review were chosen based on the PRISMA (preferred reporting items for systematic reviews and meta-analyses) Guidelines [18,19]. A literature search of the PubMed database for (1) “8q24 gene desert”, revealed 28 papers with 6 irrelevant to tumorigenesis (Figure 2A) and (2) “FAM84B”, identified 21 articles, including non-English papers (*n* = 1) and articles not directly related to FAM84B and cancer (*n* = 2) (Figure 2B). After excluding these items, 22 articles on 8q24 gene desert and 18 papers related to the FAM84B topic have been retrieved and discussed (Figure 2).

After reviewing FAM84B’s contributions to oncogenesis, we will propose a model to discuss FAM84B’s oncogenic roles in the context of Myc-derived tumorigenesis, i.e., a potential mechanistic pathway for which FAM84B collaborates with Myc during tumor formation.

## 2. Function of the 8q24.21 Gene Desert in Cancers

### 2.1. Association of the 8q24.21 Gene Desert with Oncogenesis

In addition to harboring multiple non-coding transcripts in the gene desert bordered by *FAM84B* and *Myc* (Figure 1), a number of single-nucleotide polymorphisms (SNPs) have been identified in the region by genome-wide association studies (GWAS). These SNPs are mainly associated with the risk of prostate cancer [20,21,22,23], breast cancer, ovarian cancer, colorectal cancer, and bladder cancer [13,14,24,25,26]. Besides these SNP variants, amplification of 8q24.21 occurs most frequently in human cancers, including ovarian [27], colorectal [28,29,30,31], breast [32,33,34,35,36], prostate [37,38,39,40,41,42,43], and others.

Accumulative evidence reveal a clear involvement of the individual lnRNAs of the 8q24.21 gene desert in tumorigenesis (Table 1), a concept that is supported by the emerging roles of lnRNAs in tumorigenesis via complex mechanisms [44,45]. Upregulations of PRNCR1 (prostate cancer non-coding RNA1) occurred in prostate cancer (PC), and precancerous lesions PINs (prostatic intraepithelial neoplasia) and knockdown of PRNCR1 reduced the survival of PC cells and the expression of androgen receptor (AR), indicating an important role of PRNCR1 in facilitating PC via AR signaling (Table 1) [46]. The pseudogene POU5F1B lies within this gene desert (Figure 1) [25] and its elevated expression was observed in PCs [47]. POU5F1B promotes gastric cancer [48] and hepatocellular carcinoma (Table 1) [49]. Prostate Cancer-Associated Transcript 1 (PCAT1) and PCAT2 are upregulated in PC [50,51,52]. PCAT1 also promotes ovarian cancer cell proliferation [53] and is associated with poor prognosis in colorectal cancer (CRC) (Table 1) [54]. Colorectal Cancer-Associated Transcript 1 (CCAT1), CCAT2, and Cancer Susceptibility 19 (CASC19) are upregulated in CRC [55,56]. Upregulations of both CCAT1 and CCAT2 predict CRC recurrence and poor overall survival (OS) (Table 1) [56]. CASC11 promotes CRC metastasis [57], gastric cancer cell proliferation [58], and esophageal carcinoma (Table 1) [59]. An upregulation of CASC21 was very recently reported in CRC, in which CASC21 stimulates CRC via the YAP1 actions (Table 1) [60].

The 8q24.21 gene desert contributes to cervical cancer as a frequent site of viral integration by human papilloma virus (HPV), and evidence in support of this concept has been briefly reviewed by Huppi et al. in 2012 [13]. Built on the seminal detection of HPV16 DNA in 61.1% (11/18) of cervical cancers in 1983 [61], it became clear that infection by HPV is the primary etiology of cervix carcinoma, particularly with the high-risk HPV types 16 and 18 [62]. Besides HPV infection, alterations in cellular oncogenic events are also required for cervical cancer [63]. The integration of both HPV16 and HPV18 at the gene desert suggests that HPV coordinately affects oncogene alterations for cervical cancer formation [13,64,65,66,67]. The integration hot spots in the gene desert include CASC8, CASC21, and POU5F1B [68,69]. Among 3667 breakpoints of HPV integration detected in cervical carcinoma (*n* = 104), cervical intraepithelial neoplasia (*n* = 26), and 5 cervical cancer cell lines, POU5F1B is the top site of integration (9.7%) [69].

### 2.2. Upregulation of Myc as A Mechanism underlying the Gene Desert-Derived Oncogenic Activities

In light of the well-established and powerful oncogenic functions of Myc, it is expected that research exploring the oncogenic impact of those non-coding genes within the gene desert (Figure 1) has been largely focused on the regulation of Myc. HPV integration in the 8q24.21 gene desert upregulates Myc [69]. CCAT2 expression is upregulated in CRC and lnRNA CCAT2 enhances Myc expression, which likely contributes to CCAT2-facilitated CRC metastasis [70].

The major mechanistic action in enhancing Myc expression is through regulation of chromatin structure. By examination of chromatin interactions using chromosome conformation capture (3C)-based technologies, the prostate, breast, and colon cancer risk regions within the 8q24.21 gene desert display long-range physical interaction with the Myc locus in a tissue-specific manner [71,72]. These non-coding risk regions contain super-enhancer elements and TCF-4 (transcription factor 4) binding sites that enhance Myc transcription [71,73]. The long-range association of these regions with the Myc locus thus stimulates Myc transcription, which is facilitated by Wnt/β-catanin signaling through TCF-4. These enhancers are functionally important. Mice deficient in an enhancer element *Myc*-335 that lies 335kb upstream of *Myc* are protected from APC (Adenomatous polyposis Coli) mutation-induced intestinal cancer [74]. Mice deficient in multiple Myc enhancers, including *Myc-196, Myc-335*, and *Myc-540,* within 538kb upstream of Myc, exhibit >50% reductions of Myc expression in colon and prostate. Importantly, these mice are more protected from APC mutation-induced intestinal cancer compared to mice deficient in only *Myc-335* [75]. Both CCAT1 and CCAT2 interact with Myc via the formation of DNA loops, which strongly enhances Myc expression in CRC [17,70,76]. Interestingly, long-range physical associations with Myc also facilitate the transcription of lnRNAs. For instance, the physical association allows the Myc enhancer to upregulate the transcription of CARLo-5, a short form of CCAT1 [77].

Variants in the 8q24.21 gene desert also display long-range association with the non-coding *PVT1* locus that lies downstream of the Myc locus and thus outside of the gene desert bordered by *FAM84B* and *Myc* (Figure 1). A prostate cancer risk variant within the gene desert was reported to facilitate *PVT1* transcription through physical association [78]. The lnRNA PVT1 plays a critical role in Myc-driven CRC. PVT1 is co-amplified with Myc in CRC. High levels of lnRNA PVT1 helps to maintain high levels of Myc protein expression in CRC, and ablation of PVT1 prevents Myc from inducing HCT116 cell-derived tumorigenesis [79]. In addition to Myc, PVT1 also activates β-catenin and Cyclin D1 [80]. The interplay between PVT1 and Myc has been intensively studied and reviewed [15].

## 3. The Contributions of FAM84B to Oncogenesis

*FAM84B* is the only second protein coding gene bordering the 1.2 Mb gene desert (Figure 1). It is an interesting disparity considering the relatively unknown status of FAM84B in tumorigenesis compared to the well-studied oncogenic functions of Myc. Nonetheless, emerging evidence suggest the need for a closer examination of FAM84B’s involvement in oncogenesis. In this section, we will review the data related to FAM84B’s roles in cancer.

### 3.1. FAM84B Facilitates Esophageal Cancer 

FAM84B plays a role in esophageal cancer. In a small cohort study (*n* = 59), increases in FAM84B expression were observed in 39 (66%) cases [81]. Amplification of the *FAM84B* gene and increases in its expression at the protein level occur in both preclinical lesions and esophageal squamous cell carcinomas (ESCC) [82,83]. Reductions in serum FAM84B protein expression predict pathological complete response (PCR) in ESCC patients treated with neoadjuvant chemoradiation [83]. Knockdown of FAM84B in two ESCC cell lines KYSE150 and TE-1 reduced their proliferation, migration, and invasion in vitro [82], and knockdown of FAM84B in ESCC CE81T/VGH cells significantly delayed xenograft growth in vivo [83]. Upregulations of FAM84B were also reported in melanoma [84]. However, the role of FAM84B in tumorigenesis may be complex. While downregulation of serum FAM84B protein is associated with PCR in ESCC treated with neoadjuvant chemoradiation, high levels of serum FAM84B mRNA were also observed in ESCC with PCR [83]. Downregulation of FAM84B was observed in gastroesophageal junction cancer cell lines and xenograft tumors [85]. Increases in the expression of lnRNA FAM84B-AS (antisense) transcribed from the antisense strand of the *FAM84B* gene were reported to reduce FAM84B expression in gastric cancer. lnRNA FAM84B-AS facilitates gastric cancer tumorigenesis and predicts poor prognosis [86].

### 3.2. FAM84B-Mediated Enhancement of Prostate Cancer

Evidence supports FAM84B-mediated promotion of prostate cancer (PC). FAM84B locus lies within a 2Mb region that is associated with PC risk [87]. We observed a significant upregulation of FAM84B expression in DU145 PC cell-derived prostate cancer stem cells (PCSCs) [12]. This observation is in accordance with a report showing that a risk region of prostate and colon cancer in the 8q24.21 desert was able to direct reporter expression in prostate luminal stem-like cells of transgenic mice and in prostate cancer stem cells [88]. PCSCs play critical roles in PC initiation and progression, including metastasis and therapy resistance [89]. PC mainly metastasizes to the bone [90]. The standard of care for metastatic PC (mPC) is androgen deprivation therapy (ADT). While the therapy shows remarkable response in more than 80% of cases, castration-resistant metastatic PCs (mCRPCs) commonly develop [91], to which effective therapy remains challenging. In this regard, bone metastasis and CRPC are considered major progression with poor prognosis. Of note, in comparison to prostate (*n* = 181), FAM84B mRNA was elevated in PC (*n* = 343) and further increased in metastasis in two populations (primary PC, *n* = 131, versus metastatic PC, *n* = 19; primary PC, *n* = 181, versus metastatic PC, *n* = 37) [12]. In vivo, FAM84B protein was expressed at higher levels in PCSCs-generated xenografts compared to non-PCSCs-produced xenografts, in lung metastasis compared to subcutaneous xenografts, and in CPRC produced in castrated prostate-specific *PTEN^-/-^* mice compared to PC generated in intact *PTEN^-/-^* mice [12]. Amplification of the *FAM84B* gene occurs more frequently in mCRPC (121/467 = 26%) compared to primary PCs (26/546 = 4.8%, *p* < 0.0001), and the amplification associates with reductions in disease-free survival (DFS) [12]. Additionally, increases in FAM84B mRNA expression contribute to the biomarker potential of a multigene panel in stratification of the risk of PC biochemical recurrence [92]. Collectively, a comprehensive set of evidence supports the association of FAM84B with PC tumorigenesis, metastasis, and CRPC development.

Functionally, FAM84B overexpression enhances DU145 cell invasion in vitro, subcutaneous xenograft tumor growth in vivo, and lung metastasis in a tail-vein mouse model [93]. In comparison to empty vector (EV) cell-produced xenograft tumors, those generated from FAM84B overexpression DU145 cells display elevations in AKT activation and reductions in BAD (BCL2 associated agonist of cell death) expression. Furthermore, RNA sequencing (RNA-seq) analysis revealed a large number of differentially expressed genes (DEGs) in DU145 FAM84B xenografts compared to DU145 EV tumors. These DEGs affect cell cycle progression, Golgi to ER (endoplasmic reticulum) process, mitochondrial events, and translation regulations [93]. A multigene signature (SigFAM) was derived from these DEGs, which robustly stratifies the risk of PC biochemical recurrence [93].

Structurally, FAM84B shares similarities with the H-Ras-like suppressor (HRASLS) family HRASLS1-5 within their LRAT (lecithin:retinal acyltransferase) homologous domain (Figure 3). A sub-region (residues 119–145) within LRAT is highly conserved between FAM84B and the HRASLS family (Figure 3). Deletion of this region abolished FAM84B’s ability to promote PC cell invasion in vitro [93]. These observations collectively support FAM84B-mediated promotion of PC.

## 4. Potential Collaboration between FAM84B and MYC during Tumorigenesis

HRASLS1-5 possess enzymatic activities: phospholipase A_1/2_ (PLA) and O-acyltransferase (AT) activities [94] with the catalytic site being formed by histidine 23 (H23), H35, and cysteine 113 (C113) (Figure 3) [94,95]. HRASLS members can suppress H-Ras-derived tumorigenesis, in which the catalytic activities play a role [94]. HRASLS1/A-C1 inhibits the proliferation of H-Ras-transformed NIH3T3 cells (Table 2) [96]. Ectopic expression of HRASLS2 suppresses the colony formation of HCT116 (colon cancer) and HeLa (cervical cancer) cells, and reduces the active Ras (Ras-GTP) and Ras expression in HtTA cervical cancer cells (Table 2) [97]. HRASLS3 (H-rev107) was most thoroughly studied in the HRASLS family for suppression of Ras activity. H-rev107 was identified for reversal of H-Ras-derived transformation of rat fibroblasts [98], the PLA/AT activities of HRASLS3 suppress H-Ras signaling [99], and HRASLS3 inhibits K-Ras signaling via a physical association (Table 2) [100]. HRASLS4 (RIG1, TIG3, RARRES3) suppresses Ras activation [101] and the lung metastasis of breast cancer (Table 2) [102].

While the mechanisms responsible for FAM84B-derived oncogenesis remain unclear, its direct association with the gene desert as the only other protein coding gene suggests that FAM84B contributes to Myc’s oncogenic actions. This possibility is consistent with the theme of 8q24.21 gene desert in facilitating Myc actions. This concept is also intriguing considering the similarities shared between FAM84B and the HRASLS family. Among the catalytic triad, FAM84B conserves H23 and H35 but not C113 (Figure 3) [93]. Unlike the HRASLS family, FAM84B does not possess PLA/AT enzymatic activities and will not suppress Ras signaling. To the contrary, FAM84B displays oncogenic activities. With this knowledge, it is an interesting scenario for FAM84B to facilitate Ras signaling via inhibiting the actions of the HRASLS family, and thereby in part contributing to its collaboration with Myc. The collaboration between Myc and Ras was the first demonstration of oncogene collaboration and is the most widely studied relationship [11]. Mechanisms of this collaboration are complex, and the FAM84B concept will be a new avenue in this collaboration considering its genome proximity to the Myc gene locus. Its functional and genetic linkage with Myc would suggest a co-regulatory pattern with Myc. In support of this possibility, mice deficient in the 430kb region encompassing CCAT1, POU5F1B, CCAT2, and CASC8 within the 8q24.21 gene desert (Figure 1) downregulate both FAM84B and Myc expression in mammary gland and prostate [103]. Deletion of this region delays the growth of luminal, Her2, and basal breast cancer in *MMTV-PuVT*, *MMTV-Neu*, and *C3(1)-TAg* transgenic mouse models for breast cancer, respectively [103]. Furthermore, among 35 cancer types within the cBioPortal database, 20 cancer types show co-amplification of FAM84B and Myc with the rate ≥5%. Ovarian cancer and breast cancer are the first (>40%) and fifth (>20%) cancer type with respect to the prevalence of FAM84B and Myc co-amplification [103]. This co-amplification associates with poor overall survival in breast cancer [103].

The co-amplification has also been detected in acute myeloid leukemia [104], esophageal cancer [105], colorectal carcinoma [106], and prostate cancer [103]. For prostate cancer (PC), we have analyzed gene amplification for Myc and FAM84B in all independent published datasets (*n* = 12) containing 3546 patients within cBioPortal (http://www.cbioportal.org/) [107,108] (Figure 4A). Among PCs with *Myc* amplification, 85% of cases have *FAM84B* concurrently amplified, and among tumors with *FAM84B* amplification, 96.8% of cases show *Myc* co-amplification (Figure 4A). Neuroendocrine and metastatic PCs are well-known for having poor prognosis [89]. Of note, 55.5% (152/274) of PCs with the co-amplification are aggressive PC types (metastatic and neuroendocrine PCs) (Figure 4A). In line with this evidence, PCs with FAM84B and Myc co-amplification are associated with reductions in overall survival (Figure 4B). Collectively, evidence supports an intriguing collaboration between FAM84B and Myc.

## 5. Conclusions

Along with Myc being the most commonly amplified gene in human cancers, the 8q24.21 gene desert bordered by both *Myc* and *FAM84B* is also frequently amplified [3]. While the oncogenic functions of the gene desert likely involve complex networks, it promotes tumorigenesis at least in part via facilitating Myc’s actions. In this regard, we proposed a collaboration between FAM84B and Myc which may involve Ras. Demonstration of this possibility is straightforward owning to the rich knowledge on the collaboration between Ras and Myc. However, the interaction between FAM84B and Myc is likely not limited to the potential connection of FAM84B and Ras. For instance, FAM84B and Myc may interact via lnRNAs within the 8q24.21 gene desert. This possibility is in accordance with the co-downregulation of both genes in mice with knockout of a 430Kb fragment within the gene desert [103]. Direct interactions between FAM84B and Myc at both the protein and transcriptional levels are also possible, and the latter is intriguing in view of Myc being a transcriptional factor [109]. Regardless of what major routes FAM84B may employ in its interaction with Myc, this interaction is certainly an appealing avenue of investigation. This proposition is based both on the accumulating evidence for an oncogenic role of FAM84B as well as the association of FAM84B and Myc with the 8q24.21 gene desert. While the impact of FAM84B on tumorigenesis has been relatively well-studied in prostate cancer, its oncogenic functions in general and its potential relationship with the HRASLS family should be explored in the future.

## Figures and Tables

**Figure 1 genes-11-00312-f001:**
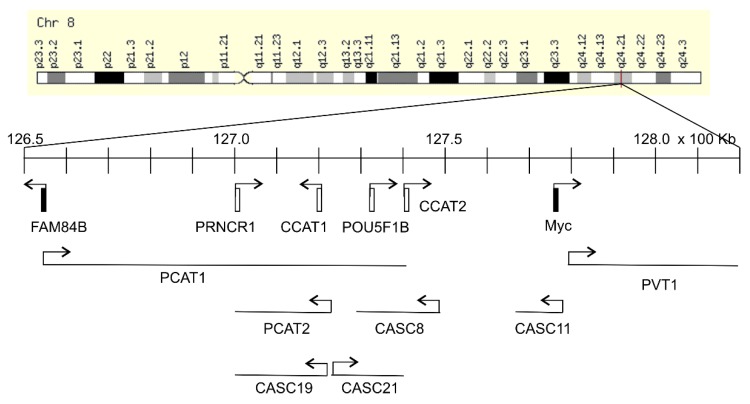
8q24.21 gene desert. The chromosome 8 image was reproduced from MYC GeneCards. The location of indicated gene and transcription direction are indicated. The gene location is defined by the Genome Reference Assembly Human Genome build 38 (GRCh38/hg38), which might be different from previous publications in which the loci of these genes were based on GRCh37/hg19 (an older version). The precise locations are LRATD2 (FAM84B, 126,552,438–126,558,478bp), PCAT1 (126,552,462–127,419,050), PCAT2 (127,072,694–127,227,541), PRNCR1 (127,079,873–127,092,600), CCAT1 (127,207,382–127,219,268), POU5F1B (127,322,183–127,420,066), CCAT2 (127,400,398–127,402,150), CASC8 (127,277,048-127,482,140), CASC11 (127,673,883-127,735,897), CASC19 (127,072,694–127,227,541), CASC21 (127,244,637–127,392,631), Myc (127,735,434–127,742,951), and PVT1 (127,794,523-128,188,211). Among these genes, only *FAM84B* and *Myc* are protein coding genes. The gene desert region is bordered by *FAM84B* and *Myc.*

**Figure 2 genes-11-00312-f002:**
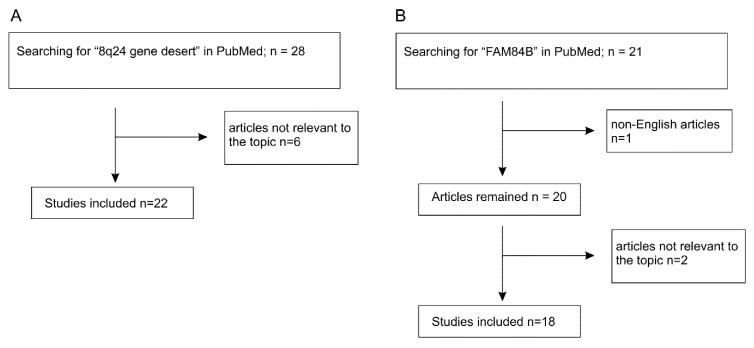
Systemic literature searching conditions and selection of articles for review.

**Figure 3 genes-11-00312-f003:**
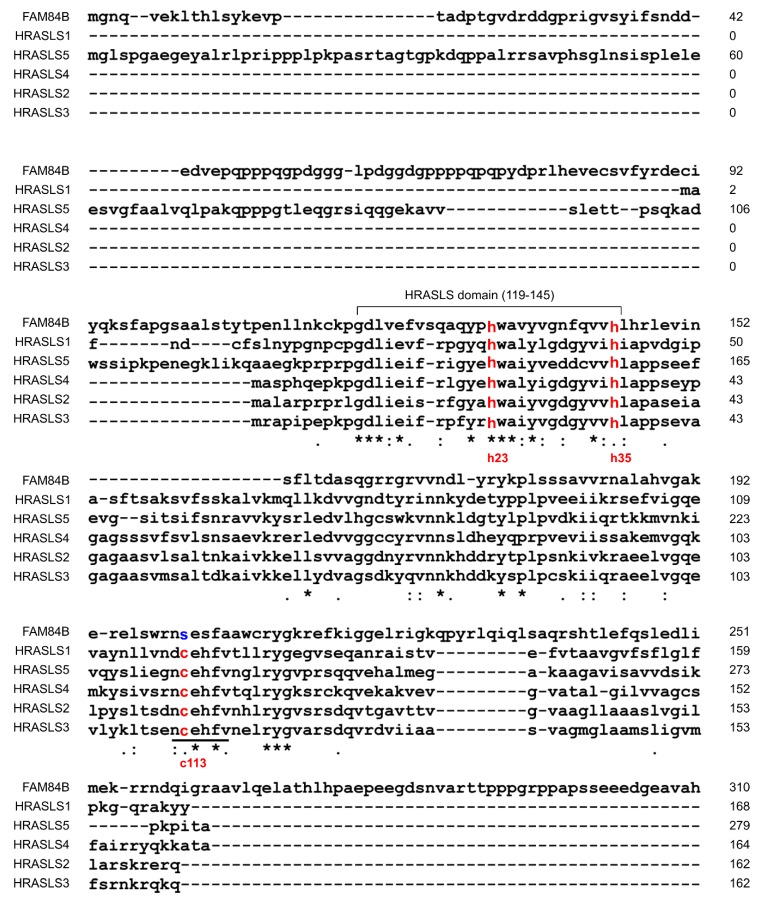
Alignment of FAM84B with the HRASLS family members. The alignment was carried out using CLUSTAL 2.1 Multiple Sequence Alignments. The three active residues (histidine 23/H23, H35, and cysteine 113/C113) are numbered based on HRASLS2 and are indicated (red). The highly conserved residues NCEHFV in HRASLS1-5 are underlined. The FAM84B fragment containing residues 119–145, which shows high homology with HRASLS1-5, is defined as HRASLS domain (119–145).

**Figure 4 genes-11-00312-f004:**
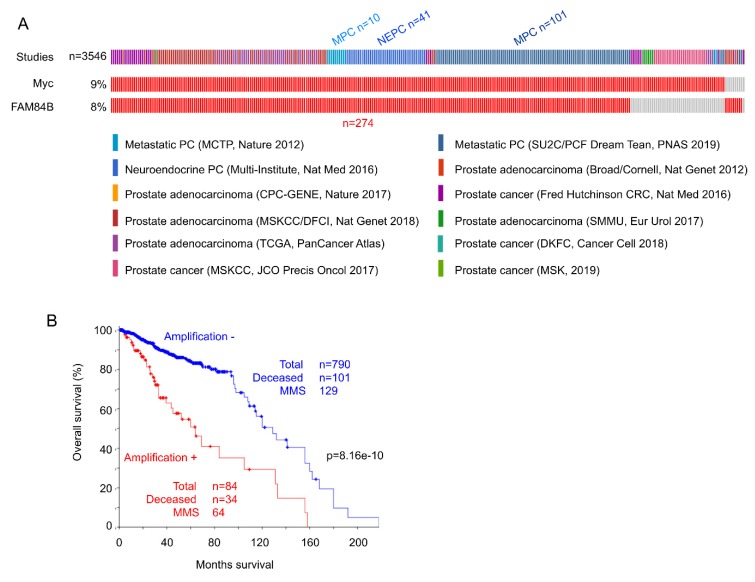
Co-amplification of *FAM84B* and *Myc* associates with poor prognosis in prostate cancer. (**A**) The 12 published studies within cBioPortal with a total number of patients *n* = 3546 were analyzed for amplification of the *Myc* and *FAM84B* genes. Individual tumors with amplification of either genes are shown, and only tumors with the indicated gene amplification are included. MPC: metastatic prostate cancer; NEPC: neuroendocrine prostate cancer. (**B**) PCs with co-amplification of *FAM84B* and *Myc* are associated with reductions in overall survival. Kaplan–Meier curve and log-rank test were performed using the program provided by cBioPortal. MMS: median months survival.

**Table 1 genes-11-00312-t001:** Association of lnRNA of the 8q24.21 gene desert with cancers.

lnRNA	PC	GC	ESC	HCC	OVC	CRC	Ref
PRNCR1	Exp + Cell prolif + AR sig +	NA	NA	NA	NA		[46]
POU5F1B	Exp +	Prom	NA	Prom	NA	NA	[47,48,49]
PCAT1	Exp +	NA	NA	NA	Cell prolif +	Poor OS	[50,51,52,53,54]
PCAT2	Exp +	NA	NA	NA	NA	NA	[50,51,52]
CCAT1	NA	NA	NA	NA	NA	Exp + Poor OS	[55,56]
CCAT2	NA	NA	NA	NA	NA	Exp + Poor OS	[55,56]
CASC11	NA	Cell prolif +	Prom			Met +	[59]
CASC19	NA	NA	NA	NA	NA	Exp +	[55,56]
CASC21	NA	NA	NA	NA	NA	Prom	[60]

PC: prostate cancer; GC: gastric cancer; ESC: esophageal cancer; HCC: hepatocellular carcinoma; OVC: ovarian cancer; CRC: colorectal cancer; NA: not available; Exp +: enhancement of expression; Cell prolif +: enhancement of cell proliferation; AR sig +: enhancement of androgen receptor signaling; Prom: promotion; OS: overall survival.

**Table 2 genes-11-00312-t002:** HRASLS (H-Ras-like suppressor) family suppresses Ras signalling.

Member	Function	Refs
HRASLS1	Inhibition of NIH3 Ras cell proliferation	[96]
HRASLS2	Reduction of Ras-GTP level Reduction of HCT116 and HeLa cell colony number Downregulation of Ras expression in HtTA cervical cancer cells	[97]
HRASLS3	Inhibition of Ras ability to transform rat fibroblasts Inhibition of Ras signalling Inhibition of K-Ras via binding to K-Ras	[98] [99] [100]
HRASLS4	Suppression of Ras activation Inhibition of breast cancer metastasis to the lung	[101] [102]

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
