# Peer review of "The Oncogenic Potential of the Centromeric Border Protein FAM84B of the 8q24.21 Gene Desert"

_genes, 2020, doi:10.3390/genes11030312_

Round 1

Reviewer 1 Report

The authors have attempted to review the roles and potential interactions between Myc and FAN84 B on chromosome 8q24.21.

Whilst the review does bring together the literature in the field, it does not discuss the role of microRNAs, or other regulatory elements in the locus.

The review does not propose how Myc and FAM84B may actually interact with each other. 

The manuscript does require significant correction of english and grammar to make it clearer.

For example:

Abstract:

The grammar in the abstract needs correcting so that it is clearer.

For example:  “ Mice deficient of a 430Kb fragment within the 1.2Mb gene desert reduce 18 breast cancer growth with concurrent downregulations of FAM84B and Myc expressions.”

Do the authors mean “ Mice deficient of a 430Kb fragment within the 1.2Mb gene desert have downregulation of FAM84B and Myc expression and reduced  breast cancer growth.” 

.

The language in the first paragraph is unnecessarily complicated.  For example, tumourigenesis is likely not to be the most complex pathological process depending upon what you study.

Page 1, L35 – genetic network should be genetic networks

The grammar needs checking throughout.  For example :–

L45 - The Myc gene resides ON (not in) chromosome 8q24.21.

L45 - “gene desert” should be “gene deserts”.

“on its downstream” should be “Downstream of Myc”.

Throughout the review, define acronyms before using them. E.g. Line 49 – lnRNA should be long non-coding RNA (lncRNA)

L52 – “an promising target” should be “a promising target”

These kind of corrections need to be made throughout the text.

Author Response

We thank reviewer #1 for the detail and insightful comments as well as the editing suggestions. Here are our point-by-point responses.

“Whilst the review does bring together the literature in the field, it does not discuss the role of microRNAs, or other regulatory elements in the locus.”

Authors' response – The reasons for not covering these non-coding RNAs here are because these have been recently reviewed. To avoid redundancy, we aim to mainly discuss the emerging oncogenic role of FAM84B in the context of the 8q24.21 gene desert. Nonetheless, we agree with the reviewer that the reasoning should be provided, which has now been outlined in this revision (lines 53-56, marked with red). We hope the reviewer agrees with the arrangement.

“The review does not propose how Myc and FAM84B may actually interact with each other.”

Authors' response – We appreciate this thoughtful comment. While we have suggested a connection between Myc and FAM84B through a potential regulation of Ras by FAM84B, we agree that the potential interaction can be better articulated. More in depth discussions have been added in this revision (lines 288-297, marked with red); this addition improves the manuscript, to which we thank the reviewer for the comment.

“The manuscript does require significant correction of english and grammar to make it clearer.

For example:

Abstract:

The grammar in the abstract needs correcting so that it is clearer.

     For example:  “ Mice deficient of a 430Kb fragment within the 1.2Mb gene desert reduce 18 breast cancer growth with concurrent downregulations of FAM84B and Myc expressions.”

     Do the authors mean “Mice deficient of a 430Kb fragment within the 1.2Mb gene desert have downregulation of FAM84B and Myc expression and reduced breast cancer growth.” ”

Authors' response – We have rearranged Abstract (lines 14 to 19) to make it flows smoothly. The sentence pointed out by the Reviewer has been accordingly modified (lines 18-19, marked with red).

“The language in the first paragraph is unnecessarily complicated.  For example, tumourigenesis is likely not to be the most complex pathological process depending upon what you study.

Page 1, L35 – genetic network should be genetic networks”

Authors' response – Revised (line 34, marked with red)

 “The grammar needs checking throughout.  For example :– L45 - The Myc gene resides ON (not in) chromosome 8q24.21.”

Authors' response – Corrected (line 45, marked with red)

“L45 - “gene desert” should be “gene deserts”.“on its downstream” should be “Downstream of Myc”.”

Authors' response – Revised (lines 45 and 46, marked with red).

“Throughout the review, define acronyms before using them. E.g. Line 49 – lnRNA should be long non-coding RNA (lncRNA)

Authors' response – Thanks for the comment. We have defined lnRNA at its first appearance in this revision (lines 46-47, marked with red). An effort has been made in this revision to define all abbreviations at their first appearance.

L52 – “an promising target” should be “a promising target”. These kind of corrections need to be made throughout the text.”

Authors' response – Corrected (line 52, marked with red). We appreciate the Reviewer’s comment and have thoroughly corrected this type of grammar errors throughout the manuscript. We trust the reviewer will see our effort.

Reviewer 2 Report

In many ways, this is a well-presented and detailed review manuscript entitled “The oncogenic potential of the centromeric border 3 protein FAM84B of the 8q24.21 gene desert”.   Considerable effort is demonstrated in this review in being thorough, and thoughtful of, in presenting current knowledge of FAM84B and of its possible collaboration with Myc, and its involvement in tumorigenesis. In describing the function of the 8q24.21 gene desert in cancer, the authors constructed an excellent table (Table 1) in which the association of lnRNA of the 8q24.21 gene desert with cancer types is presented.  In this same light, it serves this review well to have noted the upregulation of Myc as a mechanism underlying the gene desert-derived oncogenic activities.  Also, the contributions of FAM84B in esophageal cancer, it medicated enhancement of prostate cancer, are well presented and explained.

Minor:

Spelling:

Line 177:  “-mediaed” should be “-mediated”

Author Response

We appreciate the reviewer’s positive comments.

“Line 177:  “-mediaed” should be “-mediated””

Authors' response – Thanks; corrected (line 179, marked with red).